# Go Green and Recycle: Analyzing the Usage of Plastic Bags for Shopping in China

**DOI:** 10.3390/ijerph182312537

**Published:** 2021-11-28

**Authors:** Yong Li, Bairong Wang

**Affiliations:** 1School of Marxism, Shanghai Maritime University, Shanghai 201306, China; liyong@shmtu.edu.cn; 2School of Economics and Management, Shanghai Maritime University, Shanghai 201306, China

**Keywords:** old plastic bags, altruistic values, perceived plastic ban effectiveness, recycling, maximum willingness-to-pay price, China

## Abstract

The extensive usage of plastic bags has caused detrimental environmental damage, and an influx of research efforts have been undertaken to reduce people’s usage of plastic bags. However, studies regarding people’s reuse of plastic bags are still scarce. Therefore, this study is motivated to bridge this research gap by examining the determinants of old plastic bag usage and consumers’ maximum willingness to pay price for plastic carrier bags via a semi-structured online survey on a random sample of 777 Chinese consumers. Descriptive summary, KW test, and logistic regression were used to identify potential determinants and their influence on consumers’ usage of old plastic bags. The findings indicate that consumers’ age, altruistic values, and their perceived plastic ban effectiveness are positively associated with the usage of old plastic bags. Specifically, the elder and altruistic consumers who are positive for plastic ban effectiveness are more likely to reuse old plastic bags. Furthermore, when plastic carrier bags are priced at RMB 2.0, 81.2% of the surveyed consumers say they will stop buying them. Based on the pricing setting experience from the Irish plastic bag policy, we suggest that the optimal price of plastic carrier bags is RMB 12.0, which is also six times of the consumer’s maximum willingness to pay price.

## 1. Introduction

Plastic bags are widely used by modern people for shopping thanks to their desirable properties of light-weight, waterproof, low-cost, and convenience [1,2]. However, the excessive use of plastic bags and lack of reuse culture have generated noticeably negative impacts on the environment and posed serious challenges to modern solid waste management [3,4,5,6,7]. Plastic bag waste has become a major cause of soil and water contamination, as well as a blight on the landscape’s beauty [8,9]. Moreover, plastic bags are made of polyethylene, and take about 200 years or longer to naturally degrade [2,10]. However, plastic bags are used for simply 12 min on average [11]. Among the 0.5–1 billion plastic bags used globally each year, only a very small portion is recycled [4,12,13,14,15]. Rates of recycling for plastic bags are very low [16]. A survey carried out in South Africa also reported that an overwhelming majority of people do not reuse plastic bags for shopping [17].

To further improve the plastics reducing effects, extensive efforts have been made to reduce the usage of plastic bags and to promote the recycling of plastic bags. For instance, the Nepal government issued a Plastic Bag Reduction and Regulation Directive in 2011, which prohibits the usage of plastic bags which are thinner than 20 microns in the retail sector [8]. The Chinese government introduced a national plastic ban law in 2008 and required all retailers to charge their consumers for the plastic carrier bags at the point of sale [18]. Moreover, plastic bags thinner than 25 microns are banned from use and only those thicker than 25 microns are allowed for production and circulation in China [19]. The reason to increase the thickness of circulated plastic bags is to make the plastic bags durable and therefore convenient for reuse. Generally, thicker plastic bags can be reused several times. In 2020, the Chinese government upgraded the plastic ban law by requiring that by the end of 2020, all supermarkets in major cities are required to ban the usage of non-biodegradable plastic carrier bags, which are mainly made of plant based polylactic acid [20]. From 2020, the prices for carrier bags are also increased substantially to reduce people’s purchase of plastic bags and promote their reuse of old plastic bags [19].

Reusing plastic bags is perceived as one of the typical pro-environmental behaviors given the plentiful benefits this behavior could bring about, such as saving money, conserving resources, and protecting the environment. However, as most of the existing research focuses their efforts on the usage of plastic carrier bags [4,12,21], results regarding people’s reuse of old plastic bags are scarce. Then the dynamics behind this behavior become crucial to boost this recycling behavior and to reduce the extensive usage of plastic bags [21,22,23]. In this regard, this research aims to analyze the determinants of consumers’ usage of old plastic bags for shopping. As an example, people’s altruistic values could exert substantial influence on their behaviors in various aspects, such as recycling [24,25]. As another example, people’s perceived effectiveness of policies may impact their responses by forming a pressure to obey the rules and be consistent with the majority [22,26]. Therefore, this study is set to firstly test the positive impact of altruistic values on consumers’ usage of old plastic bags. Secondly, this study tests the positive impact of perceived plastic ban effectiveness on consumers’ old plastic bag usage behavior. In addition, to learn people’s tolerance for potential cost of being without bags, which may indirectly impact people’s plastic bags recycling [27], this study also investigates consumers’ maximum willingness to pay price for carrier bags. Results of this study could enrich existing literature on people’s recycling behaviors in plastics and provide valuable practical implications for reducing people’s plastic bags consumption. The study is organized as follows: the theoretical framework and research hypotheses are proposed in Section 2. Section 3 introduces the materials and methods. Section 4 and Section 5 present the results. Section 6 concludes and provides potential practical implications.

## 2. Theoretical Framework and Research Hypotheses

### 2.1. The Influence of Altruistic Values on Pro-Environmental Behavior

According to Stern [28], pro-environmental behaviors are those behaviors exerting positive effects on ecosystems. Environmentally friendly actions are beneficial to the ecosystem and may necessitate varying amounts of human sacrifices, or at the very least, do not hurt the ecosystem [29]. Therefore, to do pro-environmental behaviors, altruistic spirits are necessary. Altruistic values reflect people’s concerns for the welfare or benefits of the whole society and the environment rather than for maximizing an individual’s benefits [30,31]. Previous studies suggest that altruistic values are a powerful predictor of pro-environmental behaviors [32,33,34]. People who give the first priority to altruistic values are more likely to participate in pro-environmentally activities than those who give their first priority to egoistic values [35,36,37,38,39,40]. For instance, altruistic values are found to be effective to explain recycling, environmentally friendly traveling, purchasing environmental friendly goods, and drinking tap water [24,25]. Altruistic people are more likely to help others, sacrifice their own interest, and be more cooperative in achieving the goal of the whole society and the environment at large [34]. As recycling could generate great benefits to the whole environment, but produces few benefits or even some inconvenience to individuals, people with stronger altruistic values may respond more positively to recycling behaviors [41]. Thus, we infer that altruistic consumers are more concerned about environment protection and are therefore more likely to reuse old plastic bags for the benefits of the environment. Based on the discussions above, we propose our first hypothesis regarding people’s reuse of old plastic bags and their altruistic values as follows:

**Hypothesis** **1** **(H1).**
*Altruistic values have a positive effect on old plastic bag usage behavior.*


### 2.2. The Influence of Perceived Plastic Ban Effectiveness on Pro-Environmental Behavior

To tackle with the environment damage caused by plastic bags, the Chinese government has enacted two plastic bag ban policies in 2008 and 2020, respectively [18,19]. The new 2020 plastic ban policies forbid the usage of non-biodegradable plastic bags in supermarkets from China’s major cities [20,42]. Retailers will be fined RMB 10,000–100,000 if they violate the rules [43,44]. Existing research has shown that plastic ban is a key determinant of plastic bags usage behavior [8,45]. For instance, previous studies suggest that plastic ban significantly modifies consumers’ behaviors, such as the increased usage of reusable bags and old plastic bags [8,19,26]. It has been reported that there is a positive correlation between plastic ban and recycling behavior of plastic bags in Hong Kong [22]. As China is identified as a country with collective and Confucian cultures [41], Chinese consumers may tend to obey the plastic ban policies more willingly. Although China’s plastic ban has achieved positive results, policy execution loopholes still exist [19,26]. Different consumers may have different evaluations on the effectiveness of the plastic ban policies, and their perceived influential power of these policies may also vary. In this view, individuals who evaluate the plastic ban as effective may be more likely to perform pro-environmental behaviors. Thus, we assume consumers with a stronger sense of plastic ban effectiveness are more likely to reuse old plastic bags for shopping.

**Hypothesis** **2** **(H2).**
*Perceived plastic ban effectiveness has a positive effect on old plastic bag usage behavior.*


## 3. Materials and Methods

We conducted a semi-structured online survey on consumers from January to February 2021 to learn their old plastic bag usage behavior and maximum willingness to pay price for plastic carrier bags in China. A pilot study was conducted with 20 individuals to make sure the survey items were clearly articulated. Respondents voluntarily participated in the survey online, and a convenience sampling was adopted. A total of 801 questionnaires were obtained and 777 of them were valid. As consumers seldomly reused two or more old plastic bags during our data collection, in this study we classified consumers’ old plastic bag usage behavior into a binary situation, i.e., to use or not to use. As the high prices of plastic carrier bags may encourage people’s reuse of old plastic bags, in this study, we also investigated consumers’ maximum willingness to pay price for a plastic carrier bag. The main independent variables are measured in this study as follows.

### 3.1. Altruistic Values

According to the measurement scale from the existing study [33], we measured consumers’ altruistic values by five items, including: (a) Meaningful public service is very important to me. (b) It is important for me to contribute to the common good. (c) I empathize with other people who face difficulties. (d) I get very upset when I see other people being treated unfairly. (e) The happiness of others means a lot to me. A five-point Likert scale is used to address statements of attitude (1 = strongly disagree; 5 = strongly agree). The Cronbach’s alpha coefficient of altruistic values was calculated as 0.846, which is reliable.

### 3.2. Perceived Plastic Ban Effectiveness

We measured this variable with the statement of “Please measure the effectiveness of plastic ban policies on reducing the usage of plastic bags based on your own experiences”. A five-point Likert scale was used to address statements of attitude (1 = strongly un-obvious; 5 = strongly obvious).

### 3.3. Control Variables

In this study, we controlled four demographic variables that have been shown significantly related to consumers’ reuse of old plastic bags [19]. As our sample did not have consumers aged 50 or more, age in this study was measured in 3 categories, 1 = 18−29, 2 = 30−39, 3 = 40−50. Gender was coded as 1 for female and 0 for male. Education was measured in 3 categories, 1 = high school or lower, 2 = college degree, 3 = postgraduate degree. Income was measured in 2 categories, 1 = equal and above RMB 10,000 per month, 0 = below RMB 10,000 per month.

Statistics, like descriptive analysis, binary logistic model, and Kruskal–Wallis (KW) test [46] were used to conduct data analysis in this study. The KW test is a non-parametric analysis of variance test that predicts the probability values of features for each sub-band [47]. We estimated a logit model whose form is:(1)$$log\;it\left[\pi \left(Y=1\right)\right]=\beta_{0}+\beta_{1}X_{1}+\beta_{2}X_{2}+\beta_{3}X_{3}+\beta_{4}X_{4}+\beta_{5}X_{5}+\beta_{6}X_{6},$$
where π(Y=1) indicates the probability of carrying old plastic bag for shopping. *X*_1_ is perceived plastic ban effectiveness, $X_{2}$ is altruistic values, $X_{3}-X_{6}$ means a set of demographic factors, including age, gender, education, and income, respectively. The socio-demographic profile of the 777 investigated respondents is presented in Table 1. In this study, 60.5% of the respondents are female and others are male consumers. As for age, 60.2% of the respondents are aged between 18 and 29 years old, 27.3% are between 30 and 39 years old, and only 12.5% are aged between 40 and 50 years old. As for the education level, 6.7% of the respondents have a high school or lower degree, 44.9% of the respondents have a college degree, and 48.4% of the respondents have a postgraduate degree. In terms of income distribution, the majority (71.2%) of the respondents earn below RMB 10,000 per month.

The descriptive statistics of this research are presented in Table 2. The mean of consumers’ perceived plastic ban effectiveness is 2.88 out of 5, demonstrating that consumers place a low value on the impact of plastic ban. Thus, the plastic ban needs further publicity and stricter execution. The mean of consumers’ altruistic values is 3.93 out of 5, suggesting that, in general, the investigated consumers are mainly altruistic-oriented. As shown in Table 2, 45% of the respondents would reuse their old plastic bags for shopping, and the 55% others do not.

## 4. Results of the Binary Logistic Regression

Binary logistic model was used to find how socio-demographic characteristics, altruistic values, and perceived plastic ban effectiveness influence consumers’ reuse of old plastic bags. Based on the results of Table 2, 45% of the investigated respondents carry old plastic bags for shopping. Table 3 shows the binary logistic model results for old plastic bag usage behavior.

As shown in Table 3, age could significantly and positively influence the consumers’ old plastic bag usage behavior. Specifically, older consumers use old plastic bags for shopping 1.195 times more often than young consumers do. This finding is consistent with the results from Wang and Li who find that the performance of older people is more pro-environmental regarding reusing old plastic bags [19]. The age variable comes out as barely significant, but that is because almost everyone in the sample is young. Not a single respondent is over 50. This shows a bias in age that further analysis could resolve. In addition, this study finds no significant associations between gender, income, education, and old plastic usage behavior (see Table 3). The model results show that consumers’ altruistic values could significantly influence their old plastic bag usage behavior. As shown in Table 3, consumers with higher altruistic values are 1.298 times more likely to use old plastic bags for shopping. Thus, H1 is supported. The results of this study also show that perceived plastic ban effectiveness significantly influences consumers’ old plastic bag usage behavior. Respondents with higher perceived plastic bag effectiveness are 1.225 times more in favor of using old plastic bags for shopping. Thus, H2 is verified. However, the explanatory power of the equation is very low, which means we do not know very much about why people choose or do not choose to reuse plastic bags.

## 5. The Maximum Willingness to Pay Price for Carrier Bags

To reduce people’s consumption of plastic bags, consumers are charged for the plastic carrier bags when shopping. The policy has achieved satisfying results by reducing the usage of plastic carrier bags by 49% [18]. Another positive effect of the policy is stimulating consumers’ reuse of old plastic bags to avoid fees for plastic carrier bags. During our data collection, we asked the respondents the maximum price they were willing to pay for a plastic carrier bag. The results of distribution of this psychological price for stopping purchasing plastic carrier bags are shown in Figure 1. Around 65.4% of the consumers will stop buying plastic carrier bags when they are priced RMB 1.0. While, 81.2% of the consumers indicate that they will stop purchasing plastic carrier bags when they are priced RMB 2.0. The optimal pricing for a plastic bag depends on how effective the pricing policy is in reducing waste. Ireland enacted a plastic bag levy in 2002, which resulted in a 92% drop in plastic bag consumption, as well as reduced littering and negative landscape effects [27]. According to the lessons from the Irish plastic bag ban policies, the price for plastic carrier bags is set six times higher than the price of consumer’s maximum willingness to pay [27]. Thus, based on the findings of this study, the optimal price for plastic carrier bags is supposed to be RMB 12.0 (six times that of RMB 2.0).

In our study, the KW test was used to compare the psychological price of different groups and the results are summarized in Table 4. Among the investigated consumers, female consumers show a significantly (χ^2^ = 7.417; *p* = 0.006) higher willingness-to-pay price (RMB 1.9) for plastic carrier bags than their male counterparts (RMB 1.6) do. A possible reason behind this finding is that women rely more on plastic bags in daily life and are more willing to pay higher prices for them. In addition, consumers with a higher level of income also show a significantly (χ^2^ = 12.58; *p* = 0.000) higher willingness-to-pay price (RMB 2.2) than low-income consumers (RMB 1.6) do. Both age and education show no significant correlation with the psychological price of plastic bags. To sum up, female consumers and higher income consumers have higher willingness-to-pay prices for plastic bags.

## 6. Conclusions

This research employed a logistic regression model to investigate potential determinants of consumers’ old plastic bag usage and the KW test to explore the group difference in the price of consumer’s maximum willingness-to-pay price for plastic shopping bags in China. This study contributes to the literature of people’s recycling behaviors in plastics and provides practical implications on anti-plastics management. Theoretically, existing research regarding people’s reuse of old plastic bags is scarce. This study bridges the research gap and reveals the influential factors that affect the reuse of old plastic bags. The key findings of our results are listed as follows:

First, we find that altruistic values have a direct positive influence on old plastic bag usage behavior in China. Individuals with strong altruistic values are more concerned about the environment’s well-being. That is to say, altruism can stimulate people’s pro-environmental behaviors in China, which echoes the findings in Western countries [31,34,48]. Our finding confirms the universal influence of altruistic values given its wide effects in countries with different cultural backgrounds. Meanwhile, the finding adds evidence on the importance of environmental education. For example, for those with low altruistic values, an educational tool is necessary to emphasize the environmental damage caused by plastic bags both in schools and communities. Namely, raising people’s environmental knowledge and awareness could be an offset for the low level or lack of altruistic values.

Second, the study observes that the perceived plastic ban effectiveness is positively associated with old plastic bag usage behavior in China. Our results suggest that plastic ban policies have significantly altered consumer’s behavior by encouraging their reuse of old plastic bags and by gradually forming the culture of recycling, which is consistent with the previous studies [8,19,22,26]. In addition, the respondent consumers’ perceived plastic ban effectiveness is 2.88 out of 5, which is far from satisfying. For plastic crisis management, to achieve better plastic reduction and recycling effects, it is necessary to strengthen the propaganda and broadcasting of plastic ban policies. In addition, the effectiveness of the plastic ban depends on their implementation and sanctioning system, which also needs further improvement.

Third, our results reveal that 81.2% of the consumers will stop purchasing plastic carrier bags when they are priced RMB 2.0, and regarding the socio-economic aspect, female or higher income consumers are found to have higher willingness-to-pay prices for plastic bags. Based on the experience of Irish plastic bag policies as mentioned in Section 5, the optimal price of plastic carrier bags is set to be RMB 12.0, which is six times the price of the consumer’s maximum willingness to pay [27]. The finding could provide reference for future plastic management policy design. For instance, if plastic carrier bags are explicitly charged at RMB 12.0, plastic bags usage is estimated to decline by much more than 81.2%. In addition, a highly potential consequence of this pricing may be the increased reuse of old plastic bags. These findings provide insights into plastic bag levy policies in China.

Finally, in addition to implementing a complete ban of all types of plastic bags, reusing and recycling plastic bags is a feasible measure for the transition period and should be strongly promoted. For the present, plastics ban policies in China are mainly penalty-oriented [26]. Incentive-oriented measures for bag reuse are probably another way to reduce plastics and deserve more research efforts. The study suggests that the government could take measures to foster consumers to carry old plastic bags for shopping and reducing of unnecessary consumption of plastic bags by executing incentive-oriented policies. For instance, the consumers bringing their own bags for shopping can be rewarded with a discount or subsidy. As for consumers, it is necessary to develop a ‘bring your own bags for shopping’ habit to contribute to a more sustainable future.

The study is limited by the sample of 777 respondents and the research results cannot be representative of the whole old plastic usage behavior status quo in China. This study is also limited by the lack of consumers aged 50 or more, which makes our conclusions susceptible to biases. Further studies are encouraged to cover old consumers. Additionally, studies in the future are encouraged to examine the effects of more cultural and psychological variables (e.g., environmental concern, social pressure, and collectivist values) on people’s usage of old plastic bags and reusable bags.

## Figures and Tables

**Figure 1 ijerph-18-12537-f001:**
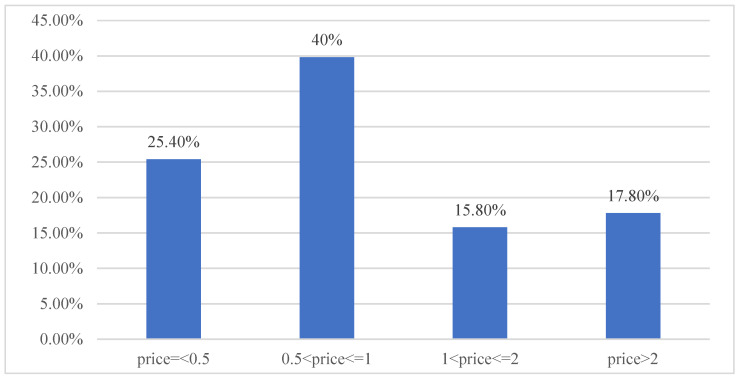
Distribution of psychological price for stopping purchasing plastic carrier bags.

**Table 1 ijerph-18-12537-t001:** Demographic profile of respondents.

Variable	Categories	Frequency	Percentage (%)
Age	18–29	468	60.2
	30–39	212	27.3
	40–50	97	12.5
Gender	Female	470	60.5
	Male	307	39.5
Education	High school or lower	52	6.7
	College degree	349	44.9
	Postgraduate degree	376	48.4
Income	Below RMB 10,000	553	71.2
	Equal and above RMB 10,000	224	28.8

Note: *N* = 777.

**Table 2 ijerph-18-12537-t002:** Descriptive statistics for the variables in the study.

Variables	Mean	SD	Min	Max
Age	1.52	0.71	1	3
Gender	0.60	0.49	0	1
Education	2.42	0.61	1	3
Income	0.29	0.45	0	1
Perceived plastic ban effectiveness	2.88	0.95	1	5
Altruistic values	3.93	0.74	1	5
Carrying old plastic bag for shopping	0.45	0.50	0	1

Note: *N* = 777.

**Table 3 ijerph-18-12537-t003:** Estimated Logit model for people’s old plastic bag usage behavior.

Variables	Estimated Coefficient	Standard Error	Significant Level	Exp(B)
Age	0.178	0.107	0.096 *	1.195
Gender	0.166	0.153	0.277	1.181
Education	0.145	0.124	0.242	1.157
Income	−0.006	0.173	0.975	0.995
Perceived plastic ban effectiveness	0.203	0.078	0.009 ***	1.225
Altruistic values	0.261	0.102	0.010 **	1.298
Constant	−2.530	0.576	0.000 ***	0.080
LR chi2	20.230	Cox and Snell R Square		0.026
−2 Log Likelihood	1048.875	Nagelkerke R Square		0.034

Notes: *N* = 777. * *p* < 0.1. ** *p* < 0.05. *** *p* < 0.01. The dependent variable is binary, with 0 = do not use old bags and 1 = use old bags.

**Table 4 ijerph-18-12537-t004:** The Kruskal–Wallis test results for the demographic factors on psychological price for stopping purchasing plastic carrier bags.

Category	Factor	Price
Age	18–29	1.6
	30–39	2.2
	40–50	1.5
	KW test	χ^2^ = 8.113
		*p* = 0.150
Gender	Female	1.9
	Male	1.6
	KW test	χ^2^ = 7.417
		*p* = 0.006
Education	High school or lower	1.5
	College degree	1.8
	Postgraduate degree	1.8
	KW test	χ^2^ = 1.978
		*p* = 0.372
Income level	High income	2.2
	Low income	1.6
	KW test	χ^2^ = 12.58
		*p* = 0.000

Note: *N* = 777.

## Data Availability

The data that support the findings of this study is available upon request to the corresponding author.

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
