# Peer review of "Go Green and Recycle: Analyzing the Usage of Plastic Bags for Shopping in China"

_ijerph, 2021, doi:10.3390/ijerph182312537_

Round 1

Reviewer 1 Report

The authors state up front that the motivation for the study is the increasing prevalence of plastic bags, which are now ubiquitous and, although many times convenient, have a tendency to cause harm to the environment.  I think this is an important topic.

In section 2 the authors lay out their research hypotheses, along with a brief account of their theoretical framework.

In section 3 they describe their data and methodology.  These sections are laid out well.  The authors make good use of subheadings, paragraph breaks and two tables.  The results of a binary regression are in Table 3. Presumably, the dependent variable is binary, with 0= do not use old bags and 1= use old bags.  But this information is not in the table.  All the independent variables are listed.  I think that between the place where the heading for the table is and the independent variables are listed along with their results, the authors should write out the information on the dependent variable (re-use = 1, do not re-use – 0).  Or the authors could place this information at the bottom, where they give the N = 777 and the p values.  This appears to be the last variable presented in Table 2, which the authors call “carrying old plastic bag for shopping.”  It looks like 45 % of the respondents do this.  I do not think it is ever appropriate to present a table on a regression without a description of the dependent variable included there, even if it was already listed in the descriptive statistics.  You have the independent variables in more than one place, why not the dependent variable?  It is actually more important than any other variable.  It is what you are trying to explain in the regression.

Although three variables emerge from this first regression as statistically significant, the explanatory power of the equation is very low.  The authors need to make clear that this means we do not know very much about why people choose or do not choose to re-use plastic bags.  The age variable comes out as barely significant, but that is because almost everyone in the sample is young.  Not a single respondent is over 50.  This shows a bias in age.  Also, 60% of respondents are below 30.  This is not representative of any community I am aware of.  In lines 190 and 191 the authors fail to mention there is no association between education and bag re-use also.  Why neglect that?

The authors correctly conclude here that both hypotheses 1 and 2 are supported by the regression results.

The authors then shift to willingness to pay for plastic bags.  I commend them for including this. The authors apparently asked an open-ended question on willingness to pay for plastic bags and then assembled the responses into four categories which they present as a histogram in Figure 1.

They site a study from Ireland that says optimal pricing for a public good is six times the maximum willingness to pay.  But the optimal pricing for an item that you are trying to reduce is not based on that, it depends on how effective the policy is in reducing waste or generating revenue to clean up the waste.  Perhaps the Irish study obtained its estimate of six times the maximum based on those considerations.

The authors descriptions on the findings associated with willingness to pay (women would pay more, as would higher income respondents) are typical in the willingness to pay literature.

The authors’ summary is essentially sound.

This was an interesting study, even though it has a few weaknesses.  It is hard to find any manuscript that doesn’t have at least some weaknesses.  I am just asking the authors to consider some of my comments I have made if they and the editor agree to making minor revisions.

Decision: Accept with minor revisions

Reviewer 2 Report

I appreciate the opportunity to review this manuscript. Overall, I find the manuscript to be more akin to a research note than a full empirical research, given the lack of theoretical contribution. That is not to say the topic is not important. The impact of single-use plastic has been profound, and it is essential that there is more deterrence on using them. Hence, many governments have started to implement various restrictions on their usage. Nevertheless, the manuscript has not sufficiently articulated the need for research and the originality of this research. The only attempt at developing the research gap can be found on page 2, between lines 59 and 60. It is only one sentence, and the sentence is not supported by any literature. The authors would need to develop a better research rationale and its contribution.

Conceptually, the study is simple and straightforward. In other words, the findings of the study are predictable and offer limited novelty. Also, the conceptual framework offers limited new insights and any meaningful contribution. Section 2.1, the authors cited Stern’s (2000) article that presented the value-belief-norm theory (by the way, the citation is not present in the reference list), but only chooses to discuss one small variable that is altruistic value. Altruistic value is a well-studied variable in pro-environmental and recycling contexts. Thus, I do not see many implications from the study.

There are also minor recommendations to the authors to help improve the manuscript. The authors may want to consider discussing the study variables in the introduction briefly and articulate why they are relevant to the study. The abstract should include more research background and not just jump directly into the aim and results. I am not sure what the authors meant about Irish on page 1, line 17. The citations and references need to be checked. Lastly, please ensure the citation format meets the journal’s requirements.

Reviewer 3 Report

The topic is of great interest. The introduction is concise and introduces the reader into the topic. Still, I recommend to the authors to include in Literature review section references to explain the importance of culture in the differences between countries regarding this topic in order to emphasize the case for China. In other words, try to respond to the question: How is culture influencing the recycling and how culture in China does that specifically. In this way, these aspects could be used by other researchers who might study the topic but in a different country. 

I suggest to rename the last sections in Conclusions. And this should include besides managerial implications, theoretical and practical ones too. Also refer to socio-economic and cultural dimensions which might influence your results. Conclusions should address also theoretical and practical implications. 

Add more recent references (2020-2021) in the literature review part and in the conclusion part you should link your results with those found in the literature. 

Reviewer 4 Report

Please find attached my remarks

Round 2

Reviewer 2 Report

I am happy to learn that the authors have kindly considered my comments from the previous round. More importantly, the authors have taken the time to revise and improve the manuscript. Overall, I find the revised manuscript to be an improvement from the original submission. However, there are still some minor issues I would like to ask the authors to address. Please see my comments below.

While I appreciate the elaboration on the Irish plastic bag policy on the abstract, however, I feel that now makes the abstract too long. Hence, the authors could easily just edit the sentence as follow ‘based on the pricing setting experience from the Irish plastic bag policy, we suggest…’ Then, the authors could remove the newly added sentence.

The second paragraph of the introduction is now very long. Would you please separate them into smaller paragraphs?

Also, the new edits in the second paragraph of the introduction did not have any literature support.

The manuscript would also benefit from a round of editing services.
